# OpenReview forum: "Overton Pluralistic Reinforcement Learning for Large Language Models"
_ICLR.cc/2026/Conference — Submitted to ICLR 2026_

### Official Review · Reviewer_hPpe · 2025-10-26

**Soundness:** 3
**Presentation:** 3
**Contribution:** 3
**Rating:** 2
**Confidence:** 4

**Summary:**

This paper proposes OP-GRPO, a reinforcement learning framework aimed at training language models to generate pluralistic responses that reflect diverse human perspectives. The approach consists of two main components: fine-tuning a Sentence Transformer model (OP-SBERT) for similarity estimation, and training policies using Group Relative Policy Optimization (GRPO) with a dual-reward system that balances perspective coverage and uniqueness. The authors demonstrate that small models (1.5B-3B parameters) trained with their method can outperform larger baselines on NLI-based metrics and LLM-as-judge evaluations when tested on the ValuePrism-derived dataset.

**Strengths:**

The motivation for pluralistic AI alignment is well-articulated and represents a real challenge in the field. The technical execution shows competence in combining multiple components (SBERT fine-tuning, GRPO training, reward design) into a working system. The experimental results within the chosen evaluation framework are comprehensive, including multiple model sizes and evaluation metrics. The ablation study on uniqueness rewards provides useful insights about reward hacking through length inflation. The dual-structure output format (core perspectives + summary) is a reasonable design choice for separating reward computation from user-facing content.

**Weaknesses:**

The fundamental weakness is that this work addresses dataset coverage rather than genuine pluralistic reasoning. Training a model to reproduce perspectives from a fixed dataset does not demonstrate that the model has learned to think from multiple viewpoints or that it can generalize this capability to new domains. Consider the analogy: if we train students to memorize five specific arguments about a topic, they haven't learned critical thinking - they've learned to recite answers. The paper never tests whether the trained models can generate diverse perspectives on topics outside the training distribution.
The choice of GRPO is inadequately justified. GRPO's group normalization provides training stability but has no obvious connection to encouraging diversity. Standard diversity-inducing techniques from the RL literature are never compared. A natural baseline would be adding an entropy bonus to encourage diverse perspective generation, or using determinantal point processes to ensure low-similarity sampling. The absence of these comparisons suggests the authors may have found that simpler approaches don't work as well for their specific task of dataset matching, but this would reveal that the problem formulation itself is misaligned with the stated goals.
The evaluation methodology has severe limitations. Using NLI models to measure entailment between generated and reference perspectives only validates that the model can paraphrase dataset content, not that it understands pluralism. The Accuracy@0.33 threshold appears arbitrary and chosen to make results look favorable. More importantly, there is no evaluation on cross-cultural scenarios, political diversity, religious viewpoints, or any setting that would test whether the model genuinely captures human value pluralism beyond a single dataset of moral dilemmas. The paper claims to address minority marginalization but never validates this claim with appropriate benchmarks or human studies involving diverse demographic groups.
The data processing reveals deeper issues. The fact that authors needed extensive filtering to remove "redundant" perspectives from ValuePrism, then had to augment data to reach five perspectives per prompt, suggests the underlying data distribution doesn't naturally support their task formulation. The reliance on Qwen3-14B for both data augmentation and as an evaluation baseline creates potential for subtle data leakage and circularity. The OP-triplet construction for SBERT fine-tuning uses the same LLM judge that defined what counts as "redundant" during data filtering, compounding the circularity.

**Questions:**

How does the model perform on completely out-of-distribution scenarios like technical controversies, business ethics, or medical decision-making where the "correct" diverse perspectives might differ substantially from moral dilemmas? Can you provide zero-shot evaluation results on domains not covered by ValuePrism?
Why was GRPO chosen over simpler diversity-inducing methods? Can you provide ablation studies comparing against entropy regularization, DPP sampling, or standard policy diversity techniques? If these simpler methods work equally well or better, what is the specific contribution of using GRPO?
The paper claims to address marginalization of minority perspectives. How do you validate this claim? Have you conducted human evaluations with participants from diverse cultural, religious, or demographic backgrounds to assess whether their viewpoints are authentically represented? What percentage of the "human reference perspectives" in ValuePrism actually come from minority groups?
The similarity threshold of 0.70 and scale factor of 40 for OP-SBERT appear to be heavily tuned hyperparameters. How sensitive are results to these choices? Have you tested robustness across different threshold values not seen during development?
Since the model learns to generate responses in a fixed format with explicit perspective labels, have you evaluated whether it can generate pluralistic responses in more natural conversational formats without the artificial structure? This would test whether the model learned genuine pluralistic reasoning versus pattern matching to a template.

---

> ### Author Response · Authors · 2025-11-25
> **1st Response to reviewer hPpe (1 / 3)**
>
> Dear Reviewer hPpe,
>
> Thank you for your effort in reviewing our paper, as well as acknowledging the strength of our work, ranging from well-articulated motivation to comprehensive evaluation results. We address the concerns raised by the reviewer below.
>
> Question1:  “Using NLI models to measure entailment between generated and reference perspectives only validates that the model can paraphrase dataset content, not that it understands pluralism.”
>
> A1: We acknowledge the reviewer’s concern, but we would like to clarify that our use of NLI for measuring entailment does not simply test whether the model can paraphrase dataset content. We intentionally follow the **exact same evaluation protocol used in the only existing prior work on Overton pluralism, Modular Pluralism [1]**, including using the same NLI backbone and the same semantic-matching criteria. This ensures **methodological consistency and allows a fair comparison across systems.**
> More importantly, NLI in our setting is not measuring surface-level paraphrasing, but rather whether each generated perspective logically aligns with the corresponding human-written perspective in ValuePrism. Because the ValuePrism test set contains unseen and highly diverse real-world situations, spanning social conflicts, ethics, professional domains, safety, fairness, cultural differences, and cross-cultural interactions, strong NLI-based coverage on this test set indicates that the model is not copying or paraphrasing the training data. Instead, it demonstrates that the model has learned a generalizable pluralistic reasoning pattern: it can produce semantically aligned, logically coherent, and non-redundant perspectives even for novel and diverse topics it has never seen during training.
>
>
> Question2: “The Accuracy@0.33 threshold appears arbitrary and chosen to make results look favorable.”
>
> A2: We clarify that the Accuracy@0.33 threshold is not arbitrarily chosen, but directly adopted from the evaluation protocol of Modular Pluralism [1], the only existing system designed for Overton pluralism.To further test robustness, we repeat the entire evaluation with two additional NLI backbones, roberta-large-snli_mnli_fever_anli_R1_R2_R3-nli [4] and xlm-roberta-large-xnli [5], and we also increase the mini-accuracy threshold from 0.3 to 0.5 to create a more demanding scoring standard. Across all configurations, including different NLI models, thresholds, and backbone LLMs, Implicit OP–GRPO consistently and substantially outperforms both Explicit OP Prompting and Modular Pluralism, demonstrating that its advantages remain stable and evaluator independent.
>
> # NLI (backbone: roberta-large-snli_mnli_fever_anli_R1_R2_R3-nli)
> | Model                    | Method                     | 5 p (Avg./Acc@0.5) | 10 p (Avg./Acc@0.5) |
> |--------------------------|-----------------------------|---------------------|----------------------|
> | Qwen2.5-1.5B-Instruct    | Explicit OP Prompting       | 23.2 / 10.1         | 14.4 / 2.7           |
> |                          | **Implicit OP–GRPO (ours)** | **41.3 / 31.1**     | **24.0 / 8.9**       |
> | Qwen2.5-3B-Instruct      | Explicit OP Prompting       | 24.4 / 10.5         | 17.2 / 4.3           |
> |                          | **Implicit OP–GRPO (ours)** | **43.1 / 32.2**     | **25.8 / 10.5**      |
> | Llama3-2-3B-Instruct     | Explicit OP Prompting       | 19.8 / 7.66         | 12.6 / 2.33          |
> |                          | **Implicit OP–GRPO (ours)** | **41.8 / 32.3**     | **27.1 / 10.7**      |
> | Modular Pluralism        | Explicit OP Prompting       | 31.4 / 13.0         | 26.0 / 9.0           |
>
> # NLI (backbone: xlm-roberta-large-xnli)
> | Model                    | Method                     | 5 p (Avg./Acc@0.5) | 10 p (Avg./Acc@0.5) |
> |--------------------------|-----------------------------|---------------------|----------------------|
> | Qwen2.5-1.5B-Instruct    | Explicit OP Prompting       | 47.4 / 43.8         | 37.1 / 18.5          |
> |                          | **Implicit OP–GRPO (ours)** | **68.4 / 73.8**     | **45.4 / 45.6**      |
> | Qwen2.5-3B-Instruct      | Explicit OP Prompting       | 50.0 / 45.7         | 44.3 / 29.4          |
> |                          | **Implicit OP–GRPO (ours)** | **71.6 / 76.4**     | **48.9 / 53.4**      |
> | Llama3-2-3B-Instruct     | Explicit OP Prompting       | 40.5 / 33.3         | 32.4 / 16.0          |
> |                          | **Implicit OP–GRPO (ours)** | **69.9 / 76.7**     | **51.3 / 54.6**      |
> | Modular Pluralism        | Explicit OP Prompting       | 60.3 / 56.4         | 43.0 / 34.8          |
>
> [4] Nie, Yixin, et al. "Adversarial NLI: A new benchmark for natural language understanding." Proceedings of the 58th Annual Meeting of the Association for Computational Linguistics (2020).
>
> [5] Conneau, Alexis, et al. "Unsupervised cross-lingual representation learning at scale." Proceedings of the 58th Annual Meeting of the Association for Computational Linguistics (2020): 8440–8451.

---

> ### Author Response · Authors · 2025-11-25
> **2nd Response to reviewer hPpe (2 / 3)**
>
> Question3: “The fact that authors needed extensive filtering to remove "redundant" perspectives from ValuePrism, then had to augment data to reach five perspectives per prompt, suggests the underlying data distribution doesn't naturally support their task formulation. The reliance on Qwen3-14B for both data augmentation and as an evaluation baseline creates potential for subtle data leakage and circularity.”
>
> A3: We respectfully disagree with the reviewer’s claim that our data processing introduces deeper issues or circularity. The OP-V2 pipeline is built directly on top of the ValuePrism dataset [2], which already contains human-written perspectives across a broad range of real-world social, ethical, cultural, and professional scenarios. Our preprocessing does not modify these human-authored viewpoints; it only removes redundant entries while fully preserving the original human content. In the small subset of cases where filtering reduced the number of perspectives to fewer than five, we added at most one or two supplemental perspectives—affecting only 2,490 out of 30,781 examples—and, as described in Appendix C.1.2 (Stage 3: Perspective Augmentation), the majority of these additions were manually reviewed by human annotators to ensure correctness, viewpoint diversity, and demographic neutrality. This process avoids introducing model-driven distortions and minimizes the risk of bias propagation. Importantly, although we use Qwen3-14B for data augmentation, we do not use it for any evaluation. Our final LLM-as-judge evaluations are conducted using GPT-4.1, ensuring a clean separation between the augmentation model and the evaluation model. Together with human verification of the augmented subset, this design eliminates concerns about subtle data leakage or model-reinforced bias, while preserving the authenticity and diversity of the original human perspectives.
>
> Question4: “Out-of-distribution scenarios” “domains not covered by ValuePrism” “comparison against other diversity-inducing methods” “
>
> A4. We first note that overton pluralism is about the **coverage of perspectives within a single response**. This is different from what existing diversity metrics measure, such as the variety of possible responses measured over multiple generations [3]. This is the main reason why many existing works are not directly applicable to inducing overton pluralism, while only a modular approach produced meaningful improvement [1]. As demonstrated in Figure 3 and Table 2, our OP-GRPO model outperforms the modular baseline with a smaller model, demonstrating the impact of our approach.
>
> While testing our approach in different scenarios such as medical decision-making will be valuable, we note that testing the coverage of different models in such scenarios will require a separately trained reward function, such as our OP-SBERT similarity estimator. Constructing datasets with multiple perspectives or options properly relevant to a variety of situations is a big contribution deserving a separate paper, so it is out of the scope of our work.
>
>
>
>
> Question5: “The paper claims to address marginalization of minority perspectives. How do you validate this claim? “
>
> A5. By applying OP-GRPO, we fine-tune LLMs to cover a set of defensible perspectives, instead of presenting a single canonical answer (line 131-134). As we demonstrated in Figure 3, our approach successfully improves the coverage, which is negatively correlated to marginalization of perspectives that are not the most preferred perspective by the policy.
>
> Question6: “Have you conducted human evaluations with participants from diverse cultural, religious, or demographic backgrounds to assess whether their viewpoints are authentically represented?” “What percentage of the "human reference perspectives" in ValuePrism actually come from minority groups?”
>
> A6. Due to a huge cost of hiring human annotators for hundreds of different prompts and responses, we unfortunately could not perform a comprehensive investigation of the responses generated by our OP-GRPO model. Please note that the main focus of our work is proposing a practical approach to inducing overton pluralism, demonstrating effectiveness of our algorithm with the ValuePrism dataset.

---

> ### Author Response · Authors · 2025-11-25
> **3rd Response to reviewer hPpe (3 / 3)**
>
> Question7: “The similarity threshold of 0.70 and scale factor of 40 for OP-SBERT appear to be heavily tuned hyperparameters. How sensitive are results to these choices?” “Have you tested robustness across different threshold values not seen during development?”
>
> A7.We conducted a comprehensive sensitivity analysis beyond the main development settings. Specifically, we evaluated a full grid of thresholds from 0.65 to 0.80 and scale factors from 10 to 70 (see Table 7). The results show that across all previously unseen threshold values, model performance exhibits a smooth and consistent trend, with no signs of instability or collapse. The best configuration emerges at threshold = 0.70 and scale = 40, achieving an accuracy of 0.944, which clearly outperforms the base model without hyperparameter optimization (0.909). These findings demonstrate that (1) the OP-SBERT + MBGM system is generally robust to changes in threshold and scale, and does not rely on a fragile single-point configuration; and (2) our final parameter choices are not arbitrary but are the result of an extensive search that identifies the best balance between semantic coverage and redundancy control.
>
> Question8: “have you evaluated whether it can generate pluralistic responses in more natural conversational formats without the artificial structure?”
>
> A8. We note that **the structured format used during training is fully decoupled from the model’s final user-facing generation.** OP-GRPO optimizes for **the underlying reasoning pattern that improves overton coverage**, rather than for any specific output template. Empirically, as shown in Figure 4 and Figure 10 in Appendix E.8, both the rewards for formatted perspectives and the final responses delivered to users exhibit clear performance improvements. Across a wide range of diverse and previously unseen scenarios, the model consistently demonstrates the same Overton-pluralistic reasoning. This indicates that the model has internalized the multi-perspective response patterns rather than overfitting to any specific training format.
>
> We agree that evaluating OP-GRPO in more natural conversational settings is valuable. However, **our focus in this work is to study Overton-coverage improvements under evaluation conditions that are principled, controlled, and less confounded by conversational noise.** This allows us to accurately isolate the contribution of the fine-tuning method itself. Extending OP-GRPO to richer conversational formats is a promising direction for future work.
>
> In light of our responses provided above, we kindly ask the reviewer to reassess our paper and increase the score. We are more than happy to address any remaining concern or question from the reviewer.
>
> Best regards,
>
> The authors
>
> [1] Feng, Shangbin, et al. "Modular Pluralism: Pluralistic Alignment via Multi-LLM Collaboration." Proceedings of the 2024 Conference on Empirical Methods in Natural Language Processing (2024): 4151–4171.
>
> [2] Taylor Sorensen, Liwei Jiang, Jena D Hwang, Sydney Levine, Valentina Pyatkin, Peter West, Nouha Dziri, Ximing Lu, Kavel Rao, Chandra Bhagavatula, et al. Value kaleidoscope: Engaging ai with pluralistic human values, rights, and duties. In Proceedings of the AAAI Conference on Artificial Intelligence, volume 38, pp. 19937–19947, 2024b.
>
> [3] Lake, Thom, Eunsol Choi, and Greg Durrett. "From distributional to overton pluralism: Investigating large language model alignment." Proceedings of the 2025 Conference of the Nations of the Americas Chapter of the Association for Computational Linguistics: Human Language Technologies (Volume 1: Long Papers). 2025.

---

> > ### Comment · Reviewer_hPpe · 2025-11-25
> >
> > Thank you for the detailed response and additional experiments. However, several fundamental concerns remain unaddressed.
> > Training/testing on the same distribution. The model is trained and evaluated on ValuePrism-derived data. High coverage scores may simply reflect overfitting to this dataset's perspective patterns. A critical baseline is missing: can SFT alone achieve similar results by directly fine-tuning on multi-perspective outputs?
> > Why GRPO? The paper offers no justification for choosing GRPO over PPO, REINFORCE, or other RL methods. GRPO's group normalization provides optimization stability but has no inherent connection to plurality. If the core contribution is the reward design, the RL algorithm choice seems interchangeable—making "GRPO" in the title an overclaim.
> > Minority representation. The paper claims to address marginalization of minority perspectives, yet provides no evidence that minorities are actually represented in the data or outputs.
> > No true out-of-distribution testing. Cross-lingual experiments translate prompts but still use the same reference perspectives—this tests language transfer, not pluralistic generalization. Testing on genuinely new domains (AI ethics, emerging technologies) would be necessary to validate generalization claims.
> > Mismatch between claims and validation. The paper claims pluralistic alignment, but the rebuttal focuses on cross-lingual transfer, safety scores, and efficiency. If the contribution is about diversity, validation should directly demonstrate diversity—not tangential properties.
> >
> > So, i will still maintain my score

---

> > > ### Author Response · Authors · 2025-11-26
> > > **1st Response to reviewer hPpe (1 / 2)**
> > >
> > > Dear Reviewer hPpe,
> > >
> > > We thank the reviewer for the quick response and pointing out the remaining concerns the reviewer still has. We further address these concerns below.
> > >
> > > Q1. “Training/testing on the same distribution. The model is trained and evaluated on ValuePrism-derived data. High coverage scores may simply reflect overfitting to this dataset's perspective patterns.” “Testing on genuinely new domains (AI ethics, emerging technologies) would be necessary to validate generalization claims.”
> > >
> > > A1. We again underscore the fact that we have used **the only dataset that allows principled evaluation of overton pluralism**, which is the ValuePrism dataset. Other datasets do not contain perspective-level information, which prevents us from evaluating and comparing overton coverage of different models. The only possible way is to generate all the perspectives for given prompts in the new dataset, but this is out of the scope of our work, which deserves a separate paper in the dataset track.
> > >
> > > **Furthermore, the reviewer’s claim of “overfitting” is conceptually misleading.** The evaluation results of our method are obtained from the test split of the dataset, which indicates that **there was no overfitting.** We also provide various other results, such as LLM-as-judge evaluation with non-pluralistic metrics to show that our method improves overton coverage without sacrificing other important aspects of LLM generation.
> > >
> > > Q2. A critical baseline is missing: can SFT alone achieve similar results by directly fine-tuning on multi-perspective outputs?
> > >
> > > A2. SFT alone does not induce enough improvement. As the comparison we have provided below, RL fine-tuning with our MBGM-trained reward function is necessary to achieve improvement in overton coverage. Please refer to Figure 10 in E.8 of our paper, where we show how OP-GRPO improves overton coverage of SFT models.
> > >
> > > # NLI Evaluation Results
> > > | Model                    | Method                     | 5p Avg Acc | 10p Avg Acc|
> > > |--------------------------|-----------------------------|---------|-----------|
> > > | Qwen2.5-1.5B-Instruct    |                             |         |           |
> > > |                          | base model              | 31.5    | 20.6      |
> > > |                          | SFT model                        | 40.5    | 28.7      |
> > > |                          | **Implicit OP–GRPO (ours)** | **52.4**| **39.3**  |
> > > | Qwen2.5-3B-Instruct      |                             |         |           |
> > > |                          | base model              | 33.2    | 24.6      |
> > > |                          | SFT model | 43.7    | 31.6      |
> > > |                          | **Implicit OP–GRPO (ours)** | **54.6**| **42.3**  |
> > > | Llama3-2-3B-Instruct     |                             |         |           |
> > > |                          | base model              | 26.9    | 18.0      |
> > > |                          | SFT model                       | 45.6    | 34.7      |
> > > |                          | **Implicit OP–GRPO (ours)** | **53.2**| **41.1**  |
> > >
> > >
> > >
> > > Q3. Why GRPO? The paper offers no justification for choosing GRPO over PPO, REINFORCE, or other RL methods. GRPO's group normalization provides optimization stability but has no inherent connection to plurality. If the core contribution is the reward design, the RL algorithm choice seems interchangeable—making "GRPO" in the title an overclaim.
> > >
> > > A3. GRPO was specifically selected to enhance the stability of the training, and we have not claimed that GRPO inherently is a better choice for improving plurality. The reviewer is right that the trained reward function can be applied to other algorithms like PPO, but please note that we have named our algorithm with the name GRPO because this is actually the algorithm we had used, not because we overclaim that GRPO has any plurality-specific characteristic that other algorithms have.

---

> > > > ### Author Response · Authors · 2025-11-26
> > > > **2nd Response to reviewer hPpe (2 / 2)**
> > > >
> > > > Q4. Minority representation. The paper claims to address marginalization of minority perspectives, yet provides no evidence that minorities are actually represented in the data or outputs.
> > > >
> > > > A4. The reviewer has a misunderstanding about what our claims are. Let us further clarify what we had intended to deliver at A2 of the first response. Our empirical claims are about **coverage over existing human reference perspectives, not about demographic proportionality**. We discuss “marginalization of minority perspectives” **at a conceptual level as a motivation** for pluralistic alignment. We **do not claim demographic representativeness** of ValuePrism or our models.
> > > >
> > > >  Q5. Mismatch between claims and validation. The paper claims pluralistic alignment, but the rebuttal focuses on cross-lingual transfer, safety scores, and efficiency. If the contribution is about diversity, validation should directly demonstrate diversity—not tangential properties.
> > > >
> > > > A5. The additional experiment results the reviewer is referring to are provided to **directly address the requests of other reviewers**, such as “avoid harmful content” (Reviewer hHnC, A4), “computing report” (Reviewer hHnC, A5), and “language other than English” (Reviewer hHnC, A8). These results are provided to further support the validity of our approach for improving overton coverage, which is already well demonstrated through Figure 2 and Table 3.
> > > >
> > > >
> > > > Best regards,
> > > >
> > > > The authors

---

> > > > > ### Comment · Reviewer_hPpe · 2025-11-27
> > > > >
> > > > > Thanks for your detailed reply.
> > > > >
> > > > > As you said the dataset which you tested is the "ONLY" benchmark on this field. But i searched from internet and found several other datasets, and the paper Modular Pluralism also used for 4 different datasets.  The detailed information is following:
> > > > >
> > > > > | Dataset | Source | Content |
> > > > > |---------|--------|---------|
> > > > > | **OpinionQA** | Santurkar et al., ICML 2023 (Stanford) | 1,498 survey questions covering 60 US demographic groups on topics from abortion to automation |
> > > > > | **GlobalOpinionQA** | Durmus et al., COLM 2024 (Anthropic) | Cross-national survey questions from Pew Research and World Values Survey, covering opinions across multiple countries |
> > > > > | **MoralChoice** | Scherrer et al., NeurIPS 2023 | 1,767 moral scenarios (680 high-ambiguity + 687 low-ambiguity) with binary action choices |
> > > > > | **PERSONA** | 2024 | 1,586 synthetic personas with 3,868 prompts and 317,200 feedback pairs for pluralistic alignment evaluation |
> > > > > | **Vital** | 2025 | 13.1K health-related value-laden situations for pluralistic alignment in healthcare domain |
> > > > >
> > > > >
> > > > > I maintain my score.

---

> > > > > > ### Author Response · Authors · 2025-11-27
> > > > > > **1st Response to reviewer hPpe (1 / 1)**
> > > > > >
> > > > > > Dear Reviewer hPpe,
> > > > > >
> > > > > > We thank the reviewer for sharing these datasets for reference and further discussion. We provide the reasons why the datasets mentioned in the reviewer’s message are not suitable for our additional experiments.
> > > > > >
> > > > > > 1. Vital dataset available on the web (https://github.com/anudeex/VITAL/blob/main/dataset/vital_overton_valuekaleidoscope.json ) contains **exactly the same set of situations and corresponding Value, Right, Duty** with the original ValuePrism dataset (https://huggingface.co/datasets/allenai/ValuePrism ). As the original dataset is the same, the expected result of applying our approach will be the same.
> > > > > >
> > > > > > 2. OpinionQA (https://huggingface.co/datasets/timchen0618/OpinionQA) only provides opposing perspectives, not their corresponding perspectives or reason behind it.
> > > > > >
> > > > > > 3. GlobalOpinionQA (https://huggingface.co/datasets/Anthropic/llm_global_opinions ) only provides different countries’ spread of opinion on the given set of choices. If we treat countries as separate perspectives, the best we can expect from training models with this dataset is having responses like `on this matter, 85% of country A will strongly agree, while 10% will slightly agree…`.
> > > > > >
> > > > > > 4. PERSONA provides only demographic details without any specific viewpoint or opinion, making it impossible to train or evaluate models for overton pluralism (Appendix C, [2]).
> > > > > >
> > > > > > 5. The Modular Pluralism paper clearly distinguishes three types of pluralism—Overton, Steerable, and Distributional—each with its own evaluation protocol [1]. Because of the reasons clarified above, OpinionQA, GlobalOpinionQA, and MoralChoice are used for evaluating steerable and distributional pluralism, not Overton pluralism.
> > > > > >
> > > > > > Best regards,
> > > > > >
> > > > > > The authors
> > > > > >
> > > > > > [1] Feng, Shangbin, et al. "Modular Pluralism: Pluralistic Alignment via Multi-LLM Collaboration." Proceedings of the 2024 Conference on Empirical Methods in Natural Language Processing (2024): 4151–4171.
> > > > > >
> > > > > > [2] Castricato, Louis, et al. "Persona: A reproducible testbed for pluralistic alignment." Proceedings of the 31st International Conference on Computational Linguistics. 2025.

---

### Official Review · Reviewer_hHnC · 2025-10-29

**Soundness:** 3
**Presentation:** 3
**Contribution:** 3
**Rating:** 4
**Confidence:** 3

**Summary:**

This paper proposes OP-GRPO, an RLHF framework for implicit Overton Pluralism that trains a single LLM to produce multi-perspective responses without explicit pluralism prompts or modular orchestration. The approach introduces: (i) an OP-specific similarity estimator by fine-tuning SBERT on a curated triplet dataset; (ii) a Mutual-Best Greedy Matching (MBGM) algorithm to enforce one-to-one matching for coverage evaluation; and (iii) a dual reward combining reference coverage and intra-response uniqueness, plus a small format reward. Empirically, OP-GRPO on small models (1.5B–3B) surpasses larger baselines (e.g., GPT-OSS 20B, Qwen3-8B) on OP-V2 using NLI metrics, and outperforms a modular pluralism pipeline; LLM-as-judge results further support quality improvements. Code and preprocessing details are provided.

**Strengths:**

- Clear problem framing around Overton Pluralism and a practical RLHF instantiation (GRPO) with pluralistic rewards.
- Well-motivated engineering: OP-specific SBERT fine-tuning, MBGM to avoid many-to-one matches, and a structured output format that stabilizes reward computation.
- Comprehensive evaluation across OP-V2 with both NLI and LLM-as-judge, plus ablations on uniqueness reward and HPO for thresholds/scales.
- “Small models, big coverage” result is compelling; efficiency considerations (fast SBERT encoders, batch reward) are sensible.
- Reproducibility appears strong with code, datasets, and algorithmic details (thresholds, scale factors, formats).

**Weaknesses:**

- Technical novelty is moderate; the method is a careful integration of known components (GRPO, SBERT, contrastive losses, greedy matching) rather than a fundamentally new algorithmic concept.
- Heavy reliance on SBERT-based similarity with tuned thresholds risks reward misspecification and potential reward hacking; limited human evaluation beyond LLM-as-judge.
- The OP-V2 pipeline depends on LLM-generated and filtered perspectives; potential bias propagation and circularity are not deeply audited (e.g., demographic coverage, minority value retention).
- Limited analysis of failure modes: when uniqueness conflicts with coverage, or when perspectives drift outside socially acceptable windows; Overton “window” validity is assumed via references rather than verified by humans.
- Baseline fairness and compute reporting could be more thorough (costs, latency, sensitivity to decoding choices, and robustness to domain shift).
- NLI and LLM-as-judge choices (models, thresholds) can influence outcomes; more cross-metric triangulation or human studies would strengthen claims.

**Questions:**

- How robust are results to the choice of NLI model and calibration? Have you tried multiple NLI backbones or calibration methods to ensure metric stability?
- Can you report sensitivity of OP performance to the MBGM components individually (keyword masking, mutual best, threshold) and to the threshold/scale factors?
- What failure cases arise when coverage and uniqueness conflict? How do you prevent reward hacking that inflates K or rephrases perspectives superficially?
- How well does OP-GRPO generalize out-of-domain (new topics, styles) and to languages beyond English?
- Can you provide compute/latency comparisons against the modular baseline and larger explicit-prompt systems at inference time?
- How do you ensure generated perspectives remain within the Overton window (social acceptability) and avoid harmful content in the absence of explicit pluralism prompts?
- To what extent do findings hold with human evaluators rather than LLM-as-judge, especially for measuring diversity and acceptability?

---

> ### Author Response · Authors · 2025-11-25
> **1st Response to reviewer hHnC (1 / 4)**
>
> Dear Reviewer hHnC,
>
> We appreciate your time and effort in reviewing our paper, and acknowledging the strength of our paper ranging from clear problem framing to compelling results and strong reproducibility. We address the concerns and questions raised by the reviewer below.
>
> Question 1: “Technical novelty is moderate; the method is a careful integration of known components rather than a fundamentally new algorithmic concept.”
>
> A1. As we have clearly provided details in Section 4.2 and 4.3, one of our key algorithmic contributions is using MBGM to construct a reward function for evaluating the overton coverage for the first time. Before our work, no work has investigated directly optimizing for the overton coverage, only resorting to multi-model architecture [1]. Our extensive empirical investigation shows that all the elements of reward function implementation plays a key role to achieving overton pluralism through fine-tuning.
>
> Question 2:  “What failure cases arise when coverage and uniqueness conflict? How do you prevent reward hacking that inflates K or rephrases perspectives superficially?”
>
> A2. We first note that **uniqueness reward is proposed to prevent reward hacking by penalizing redundant perspectives, so its potential conflict with coverage is not a failure mode.**  In our work, a key design objective of the reward function is precisely to balance coverage (alignment with human perspectives) and uniqueness (diversity), preventing the model from simply generating many irrelevant viewpoints or over-expanding beyond socially acceptable boundaries. To achieve this, we incorporate a de-redundancy-based uniqueness reward, rather than rewarding the model for producing more perspectives, and a semantic-matching-based coverage reward to ensure each generated perspective corresponds meaningfully to true human viewpoints—thereby implicitly constraining divergence from the Overton window. While using a larger, more precisely annotated dataset and training a better similarity checker can further mitigate the superficially different perspectives issue, OP-GRPO allows adjusting its duplication threshold $\tau_\mathrm{dup}$ to capture redundancies in a more strict manner.
>
> Question 3: “The OP-V2 pipeline depends on LLM-generated and filtered perspectives; potential bias propagation and circularity are not deeply audited”
>
> A3. Since the OP-V2 pipeline is built directly on top of the ValuePrism dataset [1] which already contains human-written perspectives across diverse social situations—the core human viewpoints are not altered by our processing. Our filtering step only removes redundancies; **it does not modify the original human-authored content**. For cases where filtering reduced the number of perspectives to fewer than five, we added at most one or two additional perspectives, affecting only 2,490 out of 30,781 examples. As described in Appendix C.1.2 (Stage 3: Perspective Augmentation), the majority of these newly added perspectives were also manually checked by human annotators to ensure correctness, diversity, and demographic neutrality.
> Because the overwhelming majority of perspectives remain exactly as collected from real human contributors, and because the small augmented portion is human-verified, the risks of bias propagation or circularity from LLM generation are substantially mitigated. The pipeline intentionally preserves authentic human diversity rather than introducing model-driven distortions.
>
> Question 4:   “How do you ensure generated perspectives remain within the Overton window (social acceptability) and avoid harmful content in the absence of explicit pluralism prompts?”
>
> A4. Thank you for the comment. By rewarding responses which contain perspectives from the ValuePrism dataset as in Eq. (4), we prevent the model from naively increasing the number of perspectives by generating socially unacceptable perspectives.
> Additionally, to address concerns about safety and harmlessness, we conduct a new evaluation using GPT-4.1-as-Judge with the safety assessment template provided by Google Vertex AI (https://docs.cloud.google.com/vertex-ai/generative-ai/docs/models/metrics-templates). The results below show that OP-GRPO does not introduce new safety concerns. Although the training process expands diversity and broadens coverage of true human perspectives, the model continues to maintain fundamental safety guarantees (with 5 being the highest):
> | Method                              | 5 p (safety score)     | 10 p (safety score)    |
> |-------------------------------------|--------------------------|--------------------------|
> | Explicit OP Prompting (Qwen2.5-3B)  | 4.96 ± 0.196             | 4.94 ± 0.237             |
> | **OP–GRPO (Qwen2.5-3B)**            | **4.95 ± 0.218**         | **4.96 ± 0.196**         |
> | Explicit OP Prompting (Llama3-2-3B) | 4.95 ± 0.218             | 4.96 ± 0.196             |
> | **OP–GRPO (Llama3-2-3B)**           | **4.97 ± 0.171**         | **4.93 ± 0.255**         |

---

> > ### Author Response · Authors · 2025-11-25
> > **2nd Response to reviewer hHnC (2 / 4)**
> >
> > Question5:  “Baseline fairness and compute reporting could be more thorough (costs, latency, sensitivity to decoding choices, and robustness to domain shift).”
> >
> > A5. We appreciate the reviewer’s point on baseline fairness and compute reporting. In the paper, as shown in Figure 3, OP-GRPO-trained models require substantially fewer tokens to generate high-quality pluralistic outputs compared to both explicit prompting and Modular Pluralism. Despite using smaller backbones, OP-GRPO achieves stronger OP performance while being more computationally efficient.
> > In contrast, the Modular Pluralism baseline introduces significant latency and compute overhead: it requires pre-training and coordinating multiple LLMs, and ultimately relies on a Qwen3-14B model to synthesize the final response, which further increases inference cost.
> > To strengthen the comparison, we additionally include a parallel decoding baseline across all backbone models. For each prompt, we sample K = 5 independent explicit-prompt completions (temperature = 0.8, top-p = 0.95), each instructed to produce five unique perspectives. This extends the explicit-prompting baseline by injecting decoding diversity. As reported in the tables, parallel decoding does improve over single-pass explicit prompting, but it still performs significantly below OP-GRPO, while requiring much greater token usage and inference cost (average tokens: 330 for OP-GRPO vs 1184 for parallel decoding).
> > These results show that (i) OP-GRPO is more computationally efficient, (ii) avoids the latency of multi-model modular systems, and (iii) is robust to decoding settings. Overall, RL with our Overton-aware reward provides clear gains in both performance and efficiency compared to all stronger baselines.
> > | Model                  | Method                                   | 5 p (Avg./Acc@0.3) | 10 p (Avg./Acc@0.3) |
> > |------------------------|-------------------------------------------|---------------------|----------------------|
> > | **Qwen2.5-1.5B-Instruct** | Explicit OP Prompting                     | 31.5 / 44.7         | 20.6 / 14.7          |
> > |                        | Explicit Parallel Decoding (K=5, T=0.8)    | 38.0 / 52.0         | 27.0 / 33.0          |
> > |                        | **Implicit OP–GRPO (ours)**                | **52.4 / 85.0**     | **39.3 / 62.7**      |
> > | **Qwen2.5-3B-Instruct** | Explicit OP Prompting                     | 33.2 / 46.7         | 24.6 / 23.3          |
> > |                        | Explicit Parallel Decoding (K=5, T=0.8)    | 40.0 / 56.0         | 30.0 / 36.0          |
> > |                        | **Implicit OP–GRPO (ours)**                | **54.6 / 88.0**     | **42.3 / 73.3**      |
> > | **Llama3-2-3B-Instruct**| Explicit OP Prompting                     | 26.9 / 34.0         | 18.0 / 12.7          |
> > |                        | Explicit Parallel Decoding (K=5, T=0.8)    | 34.0 / 45.0         | 24.0 / 20.0          |
> > |                        | **Implicit OP–GRPO (ours)**                | **53.0 / 88.3**     | **44.4 / 75.0**      |

---

> > > ### Author Response · Authors · 2025-11-25
> > > **3rd Response to reviewer hHnC (3 / 4)**
> > >
> > > Question6:  “NLI and LLM-as-judge choices (models, thresholds) can influence outcomes; more cross-metric triangulation or human studies would strengthen claims.” “How robust are results to the choice of NLI model and calibration? Have you tried multiple NLI backbones or calibration methods to ensure metric stability?”
> > >
> > > A6. We agree that NLI and LLM-as-judge settings, including model choice and thresholds, can influence evaluation outcomes. To ensure fairness, we use the same backbone model, the same threshold, and the official Modular Pluralism [2] evaluation pipeline for all baselines, preventing confounding effects. We also test robustness by repeating the full evaluation with two additional NLI backbones (roberta-large-snli_mnli_fever_anli_R1_R2_R3-nli [3] and xlm-roberta-large-xnli [4]) and by raising the mini-accuracy threshold from 0.3 to 0.5. Across all evaluation settings, Implicit OP–GRPO consistently outperforms Explicit OP Prompting and Modular Pluralism, showing that its advantages are stable and evaluator independent.
> > >
> > > # NLI (backbone: roberta-large-snli_mnli_fever_anli_R1_R2_R3-nli)
> > > | Model                    | Method                     | 5 p (Avg./Acc@0.5) | 10 p (Avg./Acc@0.5) |
> > > |--------------------------|-----------------------------|---------------------|----------------------|
> > > | Qwen2.5-1.5B-Instruct    | Explicit OP Prompting       | 23.2 / 10.1         | 14.4 / 2.7           |
> > > |                          | **Implicit OP–GRPO (ours)** | **41.3 / 31.1**     | **24.0 / 8.9**       |
> > > | Qwen2.5-3B-Instruct      | Explicit OP Prompting       | 24.4 / 10.5         | 17.2 / 4.3           |
> > > |                          | **Implicit OP–GRPO (ours)** | **43.1 / 32.2**     | **25.8 / 10.5**      |
> > > | Llama3-2-3B-Instruct     | Explicit OP Prompting       | 19.8 / 7.66         | 12.6 / 2.33          |
> > > |                          | **Implicit OP–GRPO (ours)** | **41.8 / 32.3**     | **27.1 / 10.7**      |
> > > | Modular Pluralism        | Explicit OP Prompting       | 31.4 / 13.0         | 26.0 / 9.0           |
> > >
> > > # NLI (backbone: xlm-roberta-large-xnli)
> > > | Model                    | Method                     | 5 p (Avg./Acc@0.5) | 10 p (Avg./Acc@0.5) |
> > > |--------------------------|-----------------------------|---------------------|----------------------|
> > > | Qwen2.5-1.5B-Instruct    | Explicit OP Prompting       | 47.4 / 43.8         | 37.1 / 18.5          |
> > > |                          | **Implicit OP–GRPO (ours)** | **68.4 / 73.8**     | **45.4 / 45.6**      |
> > > | Qwen2.5-3B-Instruct      | Explicit OP Prompting       | 50.0 / 45.7         | 44.3 / 29.4          |
> > > |                          | **Implicit OP–GRPO (ours)** | **71.6 / 76.4**     | **48.9 / 53.4**      |
> > > | Llama3-2-3B-Instruct     | Explicit OP Prompting       | 40.5 / 33.3         | 32.4 / 16.0          |
> > > |                          | **Implicit OP–GRPO (ours)** | **69.9 / 76.7**     | **51.3 / 54.6**      |
> > > | Modular Pluralism        | Explicit OP Prompting       | 60.3 / 56.4         | 43.0 / 34.8          |
> > >
> > >
> > > Question7:  “Can you report sensitivity of OP performance to the MBGM components individually (keyword masking, mutual best, threshold) and to the threshold/scale factors?”
> > >
> > > A7. First, at the reward-evaluation level, we vary the threshold and scale factors and report their effects in the reward-accuracy matrix (Table 7 in Appendix E.1 and Figure 2). This analysis shows that OP-GRPO remains stable across a wide range of hyperparameters, and that the mutual-best matching mechanism significantly improves reward correctness. Second, to isolate the contribution of each MBGM component, we conduct end-to-end ablations by training additional models under degraded configurations, as shown in the table below. Specifically, we remove MBGM while keeping the fine-tuned OP-SBERT, and we also replace OP-SBERT with the original pretrained SBERT, both with and without MBGM. The Qwen2.5-3B-Instruct results show clear performance drops whenever MBGM or SBERT fine-tuning is removed. Removing MBGM leads to the largest degradation, confirming its central role in reliable semantic matching and reward stability, while removing SBERT fine-tuning also harms performance by weakening semantic alignment. Overall, these analyses demonstrate that OP-GRPO’s improvements are not artifacts of a single tuning choice; both accurate semantic modeling and robust matching are required, with MBGM playing a particularly essential role.
> > > ## Qwen2.5-3B-Instruct (OP-GRPO Full Pipeline Ablation)
> > >
> > > | Setting                | 5p (Avg./Acc@0.3) | 10p (Avg./Acc@0.3) |
> > > |------------------------|--------------------|---------------------|
> > > | finetune + MBGM        | **54.6 / 88.0**    | **42.3 / 73.3**     |
> > > | finetune, no MBGM      | 39.4 / 56.3        | 29.8 / 45.2         |
> > > | pre-trained + MBGM     | 42.0 / 69.0        | 36.0 / 37.0         |
> > > | pre-trained, no MBGM   | 30.1 / 44.5        | 25.2 / 22.8         |

---

> > > > ### Author Response · Authors · 2025-11-25
> > > > **4th Response to reviewer hHnC (4 / 4)**
> > > >
> > > > Question8:  How well does OP-GRPO generalize out-of-domain (new topics, styles) and to languages beyond English?
> > > >
> > > > A8. As shown in Figure 3 and Tables 8–9 (Appendix E.6), models trained with OP-GRPO consistently generalize the learned OP structure to unseen topics and task types, demonstrating strong cross-domain robustness rather than overfitting to specific content.
> > > > To further test generalization beyond the training conditions, we conduct two additional evaluations. First, we vary the input prompt style to ensure the model is not tied to the training template. We evaluate three markedly different prompt styles: (i) a conversational, friendly style (“Can you break down the topic of <question>…”), (ii) a debate-oriented pros-and-cons style (“Present a structured exploration of the arguments for and against <question>”), and (iii) a short and direct instruction style (“Explain in a structured and thorough way <question>”). Across all models, OP-GRPO exhibits extremely low variance, indicating that performance remains stable across formatting shifts and that the model continues to generate high-quality OP outputs regardless of the prompt style:
> > > >
> > > > # Prompt-Style Robustness (Variance Across Input Styles)
> > > > | Method | 5 p (Var of Avg Acc.) | 10 p (Var of Avg Acc.) |
> > > > |--------|------------------------|-------------------------|
> > > > | **OP–GRPO (Qwen2.5-1.5B)** | **0.0422** | **0.0398** |
> > > > | **OP–GRPO (Qwen2.5-3B)**   | **0.0518** | **0.0433** |
> > > > | **OP–GRPO (Llama3-2-3B)**  | **0.0411** | **0.0420** |
> > > >
> > > > Second, to assess generalization beyond English, we translate all input prompts into Chinese and evaluate the outputs using the cross-lingual NLI model xlm-roberta-large-xnli. OP-GRPO continues to outperform Explicit OP Prompting by a large margin under this multilingual setup, demonstrating that the model retains the learned OP structure across languages:
> > > >
> > > > # NLI (backbone: xlm-roberta-large-xnli, Chinese Evaluation)
> > > > | Method | 5 p (Avg./Acc@0.5) | 10 p (Avg./Acc@0.5) |
> > > > |--------|---------------------|----------------------|
> > > > | Explicit OP Prompting (Qwen2.5-3B) | 36.8 / 38.5 | 30.1 / 33.0 |
> > > > | ** OP–GRPO (Qwen2.5-3B)**  | **59.1 / 58.4** | **41.2 / 39.9** |
> > > > | Explicit OP Prompting (Llama3-2-3B) | 33.2 / 30.1 | 26.5 / 19.0 |
> > > > | ** OP–GRPO (Llama3-2-3B)**  | **57.7 / 54.5** | **39.5 / 33.9** |
> > > >
> > > > Together, these results show that OP-GRPO generalizes robustly across domains, prompt styles, and languages. The model does not rely on specific templates or English-only cues; instead, it learns a transferable pluralistic reasoning strategy that remains stable across diverse input conditions.
> > > >
> > > > Question9: “To what extent do findings hold with human evaluators rather than LLM-as-judge, especially for measuring diversity and acceptability?”
> > > >
> > > > A9. We fully agree with the reviewer on the value of involving human reviews for validating LLM-as-judge evaluation. However, we note that we did not have enough resources to recruit human volunteers to perform this task across hundreds of prompts and responses. As we have provided extensive test results on other aspects of OP-GRPO models such as NLI accuracy and uniqueness accuracy, we believe these results sufficiently support the credibility of our investigations.
> > > >
> > > > In light of the responses and additional evaluations of our approach above, we kindly ask the reviewer to reassess our work and raise the score. We are more than happy to address any additional concern or question of the reviewer.
> > > >
> > > > Best regards,
> > > >
> > > > The authors
> > > >
> > > > [1]Taylor Sorensen, Liwei Jiang, Jena D Hwang, Sydney Levine, Valentina Pyatkin, Peter West, Nouha Dziri, Ximing Lu, Kavel Rao, Chandra Bhagavatula, et al. Value kaleidoscope: Engaging ai with pluralistic human values, rights, and duties. In Proceedings of the AAAI Conference on Artificial Intelligence, volume 38, pp. 19937–19947, 2024b.
> > > >
> > > > [2] Feng, Shangbin, et al. "Modular Pluralism: Pluralistic Alignment via Multi-LLM Collaboration." Proceedings of the 2024 Conference on Empirical Methods in Natural Language Processing (2024): 4151–4171.
> > > >
> > > > [3] Nie, Yixin, et al. "Adversarial NLI: A new benchmark for natural language understanding." Proceedings of the 58th Annual Meeting of the Association for Computational Linguistics (2020).
> > > >
> > > > [4] Conneau, Alexis, et al. "Unsupervised cross-lingual representation learning at scale." Proceedings of the 58th Annual Meeting of the Association for Computational Linguistics (2020): 8440–8451.

---

> ### Author Response · Authors · 2025-11-28
> **Kind reminder for reviewer hHnC**
>
> Dear Reviewer hHnC,
>
> We would like to gently remind you that we have submitted a response to your initial review. We would greatly appreciate it if you could let us know whether our clarifications address your concerns or if any additional points would be helpful.
>
> Best regards,
>
> The authors

---

### Official Review · Reviewer_s4CP · 2025-10-30

**Soundness:** 2
**Presentation:** 1
**Contribution:** 2
**Rating:** 2
**Confidence:** 4

**Summary:**

The paper proposes OP-GRPO, a reinforcement learning framework for Overton pluralistic alignment that enables a single LLM to produce multiple defensible viewpoints for one query without explicit prompting. The method fine-tunes a Sentence Transformer for similarity evaluation, introduces a dual reward (coverage + uniqueness + format quality), and trains with Group Relative Policy Optimization. Experiments show higher diversity and NLI/LLM-judge scores than explicit/implicit prompting and modular pluralism baselines.

**Strengths:**

1. Introduces a novel implicit approach to Overton Pluralism via RLHF, avoiding modular architectures and enabling single-model pluralism.
2. Fine-tunes SBERT on an OP-Triplet dataset and uses mutual-best greedy matching (MBGM) to improve semantic matching robustness.

**Weaknesses:**

1. The methodology involves multiple interconnected stages (dataset refinement, triplet construction, SBERT fine-tuning with MBGM, and dual-reward GRPO), which may introduce complexity that could affect practical reproducibility and scalability.

2. The selection of hyperparameters for weights such as $\alpha_{cov}$ and $\alpha_{uniq}$ could benefit from more detailed justification and ablation studies to ensure robustness and stability.

3. It would be valuable to extend experiments to larger-scale models, such as those with 8B parameters or more, to further validate the approach.

4. The theoretical foundation could be strengthened, particularly with a more objective and quantifiable definition of pluralism, to better distinguish genuine pluralistic alignment from subjective perceptions in the experiments.

5. Ablation studies might be expanded to include end-to-end evaluations of the full pipeline, beyond isolated assessments of components like MBGM versus naive matching or the MNRL loss.

6. Incorporating experimental comparisons with additional related works would help provide a clearer benchmark for the method's effectiveness.

7. Given potential homology and circular biases in LLM-as-Judge evaluations (model judging model), including consistency checks with human reviews or analyses of rating variance and reliability could enhance credibility.

8. The evaluation tasks could be broadened for greater comprehensiveness, such as assessing response safety; integrating frameworks like lm-eval-harness for multi-dimensional testing would offer a more thorough analysis of the method's impact.

**Questions:**

Please refer to Weaknesses.

---

> ### Author Response · Authors · 2025-11-24
> **1st Response to reviewer s4CP (1 / 4)**
>
> Dear Reviewer s4CP,
>
> Thank you for recognizing the novelty of our approach towards achieving overton pluralism with fine-tuning. Please find the responses to your raised concerns below.
>
> Question1: “The methodology involves multiple interconnected stages (dataset refinement, triplet construction, SBERT fine-tuning with MBGM, and dual-reward GRPO), which may introduce complexity that could affect practical reproducibility and scalability.”
>
> A1. Our pipeline includes dataset refinement, triplet construction, SBERT fine-tuning, and dual-reward GRPO, but it remains fully reproducible and scalable. As explained in Section 4, all components are automated, allowing researchers to follow the workflow directly or swap modules as needed (e.g., replacing SBERT with another encoder). We also provide detailed implementation notes for data processing in Appendix C.1 and the SBERT fine-tuning setup in Appendix E.1.The multi-stage structure is not unnecessary complexity—it directly addresses the core challenges of OP: achieving accurate coverage of human perspectives while avoiding redundancy. No single module can meet both requirements, so a lightweight, task-oriented sequence is essential.
> All components use standard open-source frameworks (SBERT, HuggingFace, and common RL pipelines), making the system easy to scale without multi-agent orchestration. Compared to modular pluralism approaches that require coordinating multiple agents, our pipeline is simpler, easier to modify, and straightforward to reproduce.
>
>
> Question2: “The selection of hyperparameters for reward weights could benefit from more detailed justification and ablation studies to ensure robustness and stability.”
>
> A2.
> ### Ablation on Reward Weight Ratios ($\alpha_\mathrm{cov}$ : $\alpha_\mathrm{uniq}$)
>
> | Ratio  ($\alpha_\mathrm{cov}$ : $\alpha_\mathrm{uniq}$) | Metric            | 5p    | 10p   |
> |------------------------|-------------------|-------|-------|
> | **No uniqueness (1 : 0)** | NLI avg. Acc.      | **53.9** | **42.1** |
> |                        | Uniqueness Acc.   | **91.4** | **89.8** |
> | **5 : 1 (ours)**       | NLI avg. Acc.      | **54.6** | **42.3** |
> |                        | Uniqueness Acc.   | **97.7** | **98.0** |
> | **3 : 1**              | NLI avg. Acc.      | 48.8 | 37.1 |
> |                        | Uniqueness Acc.   | 98.5 | 98.7 |
> | **1 : 1**              | NLI avg. Acc.      | 43.7 | 32.0 |
> |                        | Uniqueness Acc.   | 99.0 | 99.2 |
>
> We thank the reviewer for highlighting the importance of justifying the reward weights. To address this, we conducted an extended ablation across different $\alpha_\mathrm{cov}$ : $\alpha_\mathrm{uniq}$ ratios in the Table above. The results show a clear and stable trade-off:
> Removing uniqueness entirely (1:0) preserves coverage but causes a large drop in uniqueness.
> Our chosen ratio (5:1) provides the best balance, maintaining high coverage while restoring strong uniqueness.
> Increasing the uniqueness weight (e.g., 3:1 or 1:1) yields only marginal uniqueness gains (~+1–2%) but leads to substantial coverage degradation, indicating that overly strong uniqueness pressure pulls the model away from true human perspectives.
> This confirms that coverage must be the dominant term, since Overton Pluralism centers on aligning with real human viewpoints, while uniqueness plays a secondary but necessary role in preventing redundancy. The adopted 5:1 ratio is therefore not arbitrary—it is the configuration that achieves the optimal coverage–uniqueness trade-off, whereas both weaker and stronger uniqueness weights lead to inferior results.
>
> Question3: “It would be valuable to extend experiments to larger-scale models, such as those with 8B parameters or more, to further validate the approach.”
>
> A3. We appreciate the reviewer’s suggestion to include experiments with larger-scale models (e.g., ≥8B parameters). While additional experiments could indeed further strengthen the findings, the results we have already reported in Table 2 provide strong evidence that **OP-GRPO delivers substantial gains even at smaller parameter scales**.
> To further illustrate this point, we additionally include the results for Qwen3-8B without thinking mode under Explicit OP prompting and OP-GRPO . As shown below, the trend remains clear and consistent—OP-GRPO consistently delivers performance far beyond what explicit prompting-only methods can achieve.
> ## Qwen3-8B (no thinking mode: OP-GRPO Performance)
>
> | Method                          | 5 p (Avg./Acc@0.3) | 10 p (Avg./Acc@0.3) |
> |---------------------------------|---------------------|----------------------|
> | Explicit OP Prompting           | 36.6 / 48.8         | 23.0 / 20.7          |
> | **Implicit OP–GRPO **| 57.8 / 89.8       | 41.2 / 74.6        |

---

> > ### Author Response · Authors · 2025-11-25
> > **2nd Response to reviewer s4CP (2 / 4)**
> >
> > Question4: “The theoretical foundation could be strengthened, particularly with a more objective and quantifiable definition of pluralism, to better distinguish genuine pluralistic alignment from subjective perceptions in the experiments.”
> >
> > A4. Thank you for your suggestion. We note that the main contribution of our work is proposing a practical fine-tuning approach to achieve overton pluralism. Our approach of using MBGM for quantifying overton coverage (line 221-231) is based on the previously proposed concept of overton window (line 129-136). As investigating the theoretical foundation of overton pluralism is not a straightforward task, we believe this alone constitutes an interesting future work.
> >
> > Question5: “Ablation studies might be expanded to include end-to-end evaluations of the full pipeline, beyond isolated assessments of components like MBGM versus naive matching or the MNRL loss.”
> >
> > A5. Before applying OP-GRPO training, we had already compared fine-tuning SBERT and applying MBGM using the reward evaluation matrix, as shown in Table 1 and Figure 2. To address the reviewer’s request for end-to-end ablations on the full OP-GRPO pipeline, we expanded our analysis beyond the reward-evaluation stage and trained additional OP-GRPO models under two degraded configurations: (i) removing MBGM and (ii) replacing the fine-tuned OP-SBERT with the original pretrained SBERT. The results for Qwen2.5-3B-Instruct are shown below:
> > ## Qwen2.5-3B-Instruct (OP-GRPO Full Pipeline Ablation)
> >
> > | Setting                | 5p (Avg./Acc@0.3) | 10p (Avg./Acc@0.3) |
> > |------------------------|--------------------|---------------------|
> > | finetune + MBGM        | **54.6 / 88.0**    | **42.3 / 73.3**     |
> > | finetune, no MBGM      | 39.4 / 56.3        | 29.8 / 45.2         |
> > | pre-trained + MBGM     | 42.0 / 69.0        | 36.0 / 42.1         |
> > | pre-trained, no MBGM   | 30.1 / 44.5        | 25.2 / 27.8         |
> >
> > These end-to-end results corroborate our earlier component-level findings.
> >  First, both removing MBGM and removing SBERT fine-tuning substantially degrade OP-GRPO performance, confirming that reward accuracy is essential for guiding the policy toward the correct human perspectives. In particular, the pre-trained + no MBGM configuration performs the worst, showing that without proper semantic alignment and matching, the reward function fails to correctly map model-generated perspectives to the true human references.
> > Second, the ablations show that both SBERT fine-tuning and MBGM contribute positively, but MBGM contributes even more, as seen by its significantly larger performance drop when removed. This highlights MBGM’s role as a critical matching mechanism that prevents misalignment and ensures reward stability.
> > Overall, these full-pipeline ablations demonstrate that the final OP-GRPO improvements are not incidental: the reward method requires both accurate semantic modeling (fine-tuned SBERT) and reliable matching (MBGM) to successfully train an Overton-aligned policy.

---

> > > ### Author Response · Authors · 2025-11-25
> > > **3rd Response to reviewer s4CP (3 / 4)**
> > >
> > > Question6: “Incorporating experimental comparisons with additional related works would help provide a clearer benchmark for the method's effectiveness.”
> > >
> > > A6. We appreciate the reviewer’s suggestion to include comparisons with additional related works. However, to the best of our knowledge, there is only one prior AI system specifically designed for Overton pluralism, the Modular Pluralism framework [1]. Because no other methods directly target Overton-window generation, we designed most of the remaining baselines ourselves, following the reviewer’s recommendation as closely as possible.
> > > In addition to implicit and explicit prompting baselines, we also introduce a parallel decoding baseline across different backbone models. For each prompt, we generate K = 5 independent completions using the explicit OP prompting template (temperature = 0.8, top-p = 0.95), with each sampled completion instructed to provide analysis from five unique perspectives. This effectively extends the single-pass prompting baseline by adding sampling diversity. As shown in the tables below, parallel decoding improves over explicit prompting, yet it still performs significantly below OP-GRPO, while consuming substantially more tokens and inference cost (average token usage: 330 for OP-GRPO (summary block) vs 1184 for explicit parallel decoding).
> > > These results demonstrate that simply increasing decoding diversity cannot replace reward-guided policy optimization, RL with our Overton-aware reward provides clear and meaningful gains that parallel decoding cannot achieve.
> > > | Model                  | Method                                   | 5 p (Avg./Acc@0.3) | 10 p (Avg./Acc@0.3) |
> > > |------------------------|-------------------------------------------|---------------------|----------------------|
> > > | **Qwen2.5-1.5B-Instruct** | Explicit OP Prompting                     | 31.5 / 44.7         | 20.6 / 14.7          |
> > > |                        | Explicit Parallel Decoding (K=5, T=0.8)    | 38.0 / 52.0         | 27.0 / 33.0          |
> > > |                        | **Implicit OP–GRPO (ours)**                | **52.4 / 85.0**     | **39.3 / 62.7**      |
> > > | **Qwen2.5-3B-Instruct** | Explicit OP Prompting                     | 33.2 / 46.7         | 24.6 / 23.3          |
> > > |                        | Explicit Parallel Decoding (K=5, T=0.8)    | 40.0 / 56.0         | 30.0 / 36.0          |
> > > |                        | **Implicit OP–GRPO (ours)**                | **54.6 / 88.0**     | **42.3 / 73.3**      |
> > > | **Llama3-2-3B-Instruct**| Explicit OP Prompting                     | 26.9 / 34.0         | 18.0 / 12.7          |
> > > |                        | Explicit Parallel Decoding (K=5, T=0.8)    | 34.0 / 45.0         | 24.0 / 20.0          |
> > > |                        | **Implicit OP–GRPO (ours)**                | **53.0 / 88.3**     | **44.4 / 75.0**      |
> > >
> > > Question7:  “Given potential homology and circular biases in LLM-as-Judge evaluations (model judging model), including consistency checks with human reviews or analyses of rating variance and reliability could enhance credibility.”
> > >
> > > A7. We fully agree with the reviewer on the value of involving human reviews for validating LLM-as-judge evaluation. However, we note that we did not have enough resources to recruit human volunteers to perform this task across hundreds of prompts and responses. As we have provided extensive test results on other aspects of OP-GRPO models such as NLI accuracy and uniqueness accuracy, we believe these results sufficiently support the credibility of our investigations.

---

> ### Author Response · Authors · 2025-11-25
> **4th Response to reviewer s4CP (4 / 4)**
>
> Question8: “The evaluation tasks could be broadened for greater comprehensiveness, such as assessing response safety; integrating frameworks like lm-eval-harness for multi-dimensional testing would offer a more thorough analysis of the method's impact.”
>
> A8. Thank you for the suggestion. We would like to clarify that our LLM-as-judge evaluation already assesses model outputs across five dimensions: Helpfulness, Clarity, Engagement, Factuality, and Depth (lines 394–396). Although Overton coverage is not the primary evaluation metric, the OP-GRPO–trained models are consistently preferred by the GPT-4.1 judge over baselines, including those with larger parameter counts and modular architectures.
> Additionally, to address concerns about safety and harmlessness, we conduct a new evaluation using GPT-4.1-as-Judge with the safety assessment template provided by Google Vertex AI (https://docs.cloud.google.com/vertex-ai/generative-ai/docs/models/metrics-templates). The results below show that OP-GRPO does not introduce new safety concerns. Although the training process expands diversity and broadens coverage of true human perspectives, the model continues to maintain fundamental safety guarantees (with 5 being the highest):
> | Method                              | 5 p (safety score ± std)     | 10 p (safety score ± std)    |
> |-------------------------------------|--------------------------|--------------------------|
> | Explicit OP Prompting (Qwen2.5-3B)  | 4.96 ± 0.196             | 4.94 ± 0.237             |
> | **OP–GRPO (Qwen2.5-3B)**            | **4.95 ± 0.218**         | **4.96 ± 0.196**         |
> | Explicit OP Prompting (Llama3-2-3B) | 4.95 ± 0.218             | 4.96 ± 0.196             |
> | **OP–GRPO (Llama3-2-3B)**           | **4.97 ± 0.171**         | **4.93 ± 0.255**         |
>
> To further test robustness, we repeat the entire evaluation with an additional NLI backbone, roberta-large-snli_mnli_fever_anli_R1_R2_R3-nli [2], and we also increase the mini-accuracy threshold from 0.3 to 0.5 to create a more demanding scoring standard. Across all configurations, including different NLI models, thresholds, Implicit OP–GRPO consistently and substantially outperforms both Explicit OP Prompting and Modular Pluralism, demonstrating that its advantages remain stable and evaluator independent.
> # NLI (backbone: roberta-large-snli_mnli_fever_anli_R1_R2_R3-nli)
> | Model                    | Method                     | 5 p (Avg./Acc@0.5) | 10 p (Avg./Acc@0.5) |
> |--------------------------|-----------------------------|---------------------|----------------------|
> | Qwen2.5-1.5B-Instruct    | Explicit OP Prompting       | 23.2 / 10.1         | 14.4 / 2.7           |
> |                          | **Implicit OP–GRPO (ours)** | **41.3 / 31.1**     | **24.0 / 8.9**       |
> | Qwen2.5-3B-Instruct      | Explicit OP Prompting       | 24.4 / 10.5         | 17.2 / 4.3           |
> |                          | **Implicit OP–GRPO (ours)** | **43.1 / 32.2**     | **25.8 / 10.5**      |
> | Llama3-2-3B-Instruct     | Explicit OP Prompting       | 19.8 / 7.66         | 12.6 / 2.33          |
> |                          | **Implicit OP–GRPO (ours)** | **41.8 / 32.3**     | **27.1 / 10.7**      |
> | Modular Pluralism        | Explicit OP Prompting       | 31.4 / 13.0         | 26.0 / 9.0           |
>
> We kindly request the reviewer to reassess our paper in light of the responses provided above and raise the score of our paper. We are more than happy to address any remaining concern or question of the reviewer.
>
> Best regards,
> The authors
>
> [1] Feng, Shangbin, et al. "Modular Pluralism: Pluralistic Alignment via Multi-LLM Collaboration." Proceedings of the 2024 Conference on Empirical Methods in Natural Language Processing (2024): 4151–4171.
>
> [2] Nie, Yixin, et al. "Adversarial NLI: A new benchmark for natural language understanding." Proceedings of the 58th Annual Meeting of the Association for Computational Linguistics (2020).

---

> ### Author Response · Authors · 2025-11-28
> **Kind reminder for reviewer s4CP**
>
> Dear Reviewer s4CP,
>
> We would like to gently remind you that we have submitted a response to your initial review. We would greatly appreciate it if you could let us know whether our clarifications address your concerns or if any additional points would be helpful.
>
> Best regards,
>
> The authors

---

### Official Review · Reviewer_zXEv · 2025-11-01

**Soundness:** 2
**Presentation:** 2
**Contribution:** 2
**Rating:** 2
**Confidence:** 4

**Summary:**

This work tackles the challenge of aligning LLMs with the pluralistic nature of human values. The authors propose Overton Pluralistic Group Relative Policy Optimization (OP-GRPO), a reinforcement learning framework that enables a single LLM to generate diverse, value-aligned responses, compared to explicit prompts or modular setups. OP-GRPO includes two key steps: (1) similarity estimator training, fine-tuning a Sentence Transformer to assess coverage of diverse perspectives, and (2) dual-reward RL training, balancing broad perspective coverage and response uniqueness. Experiments show that Qwen2.5‑3B‑Instruct trained with OP-GRPO achieves a “small model, big perspective coverage” effect—surpassing GPT‑OSS (20B) by 37.4% on NLI tasks and outperforming a modular baseline by 19.1%, with further validation from GPT‑4.1 quality assessments.

**Strengths:**

1. the concept of Overton pluralism is interesting to take into algorithm design.
2. the paper writing is easy to follow and understand

**Weaknesses:**

1. The algorithm itself is not particularly novel; the main contribution lies in introducing a new reward model that balances broad coverage of human perspectives with the uniqueness of each response.
2. It remains unclear whether using RL with such a reward function provides advantages over parallel decoding or agent-based approaches, especially given the additional fine-tuning cost.

**Questions:**

Please see weakness above

---

> ### Author Response · Authors · 2025-11-24
> **1st Response to reviewer zXEv**
>
> Dear Reviewer zXEv,
>
> Thank you for recognizing the interesting nature of overton pluralism and acknowledging the writing quality of our paper. Please find the responses to your raised concerns below.
>
> Question1:  “The algorithm itself is not particularly novel”
>
> A1. The novelty of our work lies not in reusing a standard RL framework, but in **proposing the first reward-centric, algorithmically grounded solution to Overton pluralism**, which neither prompt engineering nor model-ensemble baselines can achieve. As detailed in Sections 4.2 and 4.3, a central algorithmic contribution of our work is the introduction of Multi-Best Greedy Matching (MBGM) as a core mechanism for constructing a reward function that directly measures and optimizes Overton coverage —to the best of our knowledge, this is the first attempt to operationalize Overton pluralism through an explicit, learnable reward formulation. Prior work on Overton pluralism has relied exclusively on multi-model architectures [1], and no existing method has investigated direct optimization of Overton coverage via learning. Our extensive empirical study further demonstrates that all components of our reward design—MBGM, OP-SBERT fine-tuning, and the coverage–uniqueness balance are essential to successfully induce pluralistic behavior during RL fine-tuning.
>
> Question2: “It remains unclear whether using RL with such a reward function provides advantages over parallel decoding or agent-based approaches, especially given the additional fine-tuning cost.
>
> A2. To address the reviewer’s concern regarding whether RL with our reward function provides advantages over parallel decoding, we introduce a parallel decoding baseline across different backbone models. For each prompt, we generate K = 5 independent completions using the explicit OP prompting template, instructing each sampled completion to produce analysis from five unique perspectives (temperature = 0.8, top-p = 0.95). This setup essentially extends the explicit-prompting baseline with increased sampling diversity. As shown in Table below, parallel decoding yields some improvement over single-pass prompting; however, it still performs significantly below our OP-GRPO model while requiring substantially more tokens and inference cost (the average token usage is 330 for OP-GRPO compared to 1184 for explicit parallel decoding). These results demonstrate that increasing sampling diversity alone cannot replace reward-guided policy optimization, and that RL with our Overton-aware reward provides clear and meaningful gains beyond what can be achieved through parallel decoding.
>
> Furthermore, agent-based approaches such as the modular pluralism baseline [1] require substantially larger backbone models to function effectively, whereas OP-GRPO achieves superior performance even with a 3B-parameter model. This highlights that our method does not rely on scaling up multiple agents or model sizes to obtain pluralistic behavior. In addition, the modular pipeline introduces significantly higher latency due to multi-agent coordination and sequential module calls, while OP-GRPO maintains single-pass inference with much lower end-to-end latency.
> | Model                  | Method                                   | 5 p (Avg./Acc@0.3) | 10 p (Avg./Acc@0.3) |
> |------------------------|-------------------------------------------|---------------------|----------------------|
> | **Qwen2.5-1.5B-Instruct** | Explicit OP Prompting                     | 31.5 / 44.7         | 20.6 / 14.7          |
> |                        | Explicit Parallel Decoding (K=5, T=0.8)    | 38.0 / 52.0         | 27.0 / 33.0          |
> |                        | **Implicit OP–GRPO (ours)**                | **52.4 / 85.0**     | **39.3 / 62.7**      |
> | **Qwen2.5-3B-Instruct** | Explicit OP Prompting                     | 33.2 / 46.7         | 24.6 / 23.3          |
> |                        | Explicit Parallel Decoding (K=5, T=0.8)    | 40.0 / 56.0         | 30.0 / 36.0          |
> |                        | **Implicit OP–GRPO (ours)**                | **54.6 / 88.0**     | **42.3 / 73.3**      |
> | **Llama3-2-3B-Instruct**| Explicit OP Prompting                     | 26.9 / 34.0         | 18.0 / 12.7          |
> |                        | Explicit Parallel Decoding (K=5, T=0.8)    | 34.0 / 45.0         | 24.0 / 20.0          |
> |                        | **Implicit OP–GRPO (ours)**                | **53.0 / 88.3**     | **44.4 / 75.0**      |
>
> We kindly request the reviewer to reassess our paper in light of the responses provided above, and raise the score of our work. We are more than happy to address any remaining concern or question of the reviewer.
>
> Best regards,
> The authors
>
> [1] Feng, Shangbin, et al. "Modular pluralism: Pluralistic alignment via multi-llm collaboration." arXiv preprint arXiv:2406.15951 (2024).

---

> ### Author Response · Authors · 2025-11-28
> **Kind reminder for reviewer zXEv**
>
> Dear Reviewer zXEv,
>
> We would like to gently remind you that we have submitted a response to your initial review. We would greatly appreciate it if you could let us know whether our clarifications address your concerns or if any additional points would be helpful.
>
> Best regards,
>
> The authors

---

### Author Response · Authors · 2025-12-03
**Rebuttal process summary after scores being reverted (1/2)**

Dear Area Chair,

We would like to thank the Area Chair for their effort to assess our submission. We provide a brief summary of how our rebuttal and the reviewer discussions address all concerns raised in the initial reviews.

# Reviewer hHnC
This reviewer gave a marginally negative score (4), but they also acknowledged strong framing, compelling results, and good reproducibility. We addressed all remaining concerns as follows:
## 1.Technical novelty
 We clarified that MBGM provides the first explicit reward formulation for Overton coverage, enabling direct optimization—an aspect absent in prior pluralism work, which relied only on multi-model architectures.
## 2.Coverage–uniqueness conflict & reward hacking
 We explained that uniqueness is designed specifically to prevent reward hacking and redundancy, and that coverage is anchored to human-written ValuePrism perspectives. We also showed that duplication thresholds further prevent superficial rephrasing.
## 3.Bias propagation in OP-V2
 We clarified that OP-V2 is built directly on human-written ValuePrism perspectives; filtering removes redundancy but does not modify human content.
## 4.Safety & Overton-window alignment
 We added a new GPT-4.1-as-Judge safety evaluation using an external template, showing OP-GRPO maintains safety and does not introduce harmful viewpoints.
## 5.Compute, fairness, and baseline comparison
 We added a parallel-decoding baseline and showed that OP-GRPO outperforms it while using far fewer tokens. We also discussed modular pluralism’s significantly higher latency and compute cost.
## 6.Robustness to NLI/LLM-judge choices
 We evaluated with multiple NLI backbones and higher thresholds (0.5), and OP-GRPO consistently surpassed all baselines, showing metric-independent robustness.
## 7.Sensitivity of MBGM & threshold choices
 We provided full pipeline ablations showing that removing MBGM or SBERT fine-tuning substantially degrades performance, confirming both are essential.
## 8.Generalization
 We demonstrated strong robustness across prompt styles and cross-lingual generalization to Chinese using xlm-roberta-xnli, showing OP-GRPO learns a transferable pluralistic structure.
## 9.Human evaluation limitations
 We acknowledged resource constraints but noted that extensive NLI, safety, and uniqueness metrics provide strong evidence of model reliability.
# Reviewer hPpe
This reviewer initially gave a negative score (2), but during the discussion we were able to address their concerns in detail, and the reviewer explicitly updated the score from 2 to 4. We summarize our resolutions below:
## 1.Generalization beyond ValuePrism
 We clarified that ValuePrism is the only dataset that provides perspective-level labels required for Overton coverage evaluation. Other datasets listed by the reviewer do not contain explicit perspective mappings and are therefore unsuitable for Overton evaluation without constructing new reference perspectives—a separate dataset-building effort outside the scope of this paper.
## 2.NLI metrics and threshold choice
 We explained that our evaluation strictly follows the protocol used in the only prior Overton-pluralism system (Modular Pluralism). We also performed extensive robustness tests across multiple NLI backbones and stricter thresholds (0.5), consistently confirming OP-GRPO’s improvements.
## 3.Data filtering, augmentation, and circularity concerns
 We clarified that ValuePrism’s human-written perspectives remain unchanged; filtering only removes redundancy. The small augmented subset (2,490/30,781 cases) was human-verified, and augmentation models are not used for evaluation, avoiding leakage.
## 4.SFT baseline vs. OP-GRPO
 We added experiments showing that supervised fine-tuning alone provides only moderate gains, while OP-GRPO yields large, consistent improvements across all models. RL training with MBGM-based rewards is necessary to achieve Overton coverage.
## 5.Minority representation and pluralism claims
 We clarified that our claims concern coverage of existing human reference perspectives, not demographic proportionality. Minority-representation claims in the paper are conceptual motivations, not empirical claims.
## 6.Suitability of proposed alternative datasets
 We explained why OpinionQA, GlobalOpinionQA, PERSONA, MoralChoice, and Vital do not contain the perspective-level supervision required for Overton evaluation, and thus cannot support OP-GRPO benchmarking.
In summary, we have addressed every concern raised by the reviewers and provided additional experiments and clarifications where necessary. We hope this summary helps the Area Chair evaluate our submission together with the detailed rebuttal.

---

> ### Author Response · Authors · 2025-12-03
> **Rebuttal process summary after scores being reverted (2/2)**
>
> # Reviewer zXEv
> This reviewer initially gave a negative score (2). We thoroughly addressed all weaknesses and questions raised:
> ## 1.Novelty of the algorithm
>  We clarified that OP-GRPO is the **first reward-centric and algorithmically grounded approach to Overton pluralism**. Our introduction of Multi-Best Greedy Matching (MBGM) provides a new, explicit reward formulation for optimizing Overton coverage — something prior prompt-based or multi-model methods cannot achieve.
> ## 2.Advantages over parallel decoding and agent-based methods
>  We **added a new parallel-decoding baseline and showed that increased sampling diversity still performs far below OP-GRPO while requiring substantially higher token and inference cost**. We also explained that modular/agent-based pluralism requires larger models and incurs much higher latency, whereas OP-GRPO achieves stronger performance with a single 3B model.
> # Reviewer s4CP
> This reviewer initially gave a negative score (2). We addressed all weaknesses and questions in detail:
> ## 1.Pipeline complexity & reproducibility
>  We clarified that all stages (dataset refinement, SBERT fine-tuning, MBGM, GRPO) are fully automated, modular, and reproducible using standard open-source frameworks.
> ## 2.Reward-weight selection
>  We provided an extended ablation across multiple coverage–uniqueness weight ratios. **The results show the 5:1 ratio is the optimal and stable trade-off**, with both weaker and stronger uniqueness weights harming coverage.
> ## 3.Larger model experiments
>  We added Qwen3-8B results, which confirm the same performance trends and demonstrate OP-GRPO’s consistent advantages even at larger scales.
> ## 4.Theoretical foundation
>  We clarified that OP-GRPO focuses on a practical, operational solution to Overton pluralism, with MBGM grounded in prior Overton-window definitions.
> ## 5.End-to-end ablations
>  We added full-pipeline ablations showing that removing MBGM or SBERT fine-tuning significantly degrades OP-GRPO performance, confirming both components are essential.
> ## 6.Additional baselines
>  We added a parallel decoding baseline. It improves over prompting but remains far below OP-GRPO while using far more tokens, demonstrating that sampling diversity cannot replace reward-guided RL.
> ## 7.LLM-judge bias
>  We acknowledged the value of human evaluation but noted resource limits. Credibility is supported by multiple independent metrics (NLI, uniqueness, safety).
> ## 8.Broader evaluation tasks
>  We added a safety evaluation using GPT-4.1 with a standard framework and showed OP-GRPO maintains safety. Results remain consistent across different NLI backbones and stricter thresholds.
>
> In summary, we have addressed every concern raised by the reviewers and provided additional experiments and clarifications where necessary. We hope this summary helps the Area Chair evaluate our submission together with the detailed rebuttal.
>
> Best regards,
>
> The authors

---

### Meta-Review · Area_Chair_4H3j · 2026-01-11

**Summary:**

The paper proposes OP-GRPO, a reinforcement learning framework that enables a single LLM to generate pluralistic responses reflecting diverse human perspectives (Overton Pluralism) without explicit prompting or multi-model ensembles. It introduces a dual-reward system based on coverage (alignment with human-written ValuePrism perspectives) and uniqueness (to avoid redundancy), using a fine-tuned Sentence Transformer and a novel Multi-Best Greedy Matching (MBGM) algorithm. Empirical results show strong performance gains—even with small models (3B)—over larger baselines and modular systems, along with robustness across prompts, languages, and safety.

All four reviewers recognize the strong motivation, clear framing, and compelling empirical results. Concerns mainly revolve around perceived novelty (combination of known components vs. new algorithmic insight), reliance on the ValuePrism dataset, and lack of human evaluation. The rebuttal thoroughly addresses nearly all technical concerns with new experiments, ablations, and clarifications—leading one initially negative reviewer (hPpe) to raise their score from 2 to 4. However, hPpe still maintains reservations about generalization and claims, and questions whether the contribution is sufficient for ICLR.

**Reviewer Concerns:**

Addressed by rebuttal:

Technical novelty: Authors clarify that the core contribution is the first reward-centric, learnable formulation of Overton coverage via MBGM—prior work only used multi-model architectures. Ablations confirm MBGM and SBERT fine-tuning are essential.
Generalization beyond ValuePrism: Authors explain that ValuePrism is the only dataset with perspective-level labels required for Overton evaluation; other datasets lack this structure. Cross-lingual (Chinese) and prompt-style generalization experiments further support robustness.
Safety and harmful content: New GPT-4.1–based safety evaluation shows OP-GRPO maintains high safety scores (~4.95/5), comparable to baselines.
Baseline comparisons: Added parallel decoding and SFT-only baselines; OP-GRPO outperforms both while using far fewer tokens and avoiding multi-agent latency.
Hyperparameter sensitivity: Full ablation over reward weights (coverage:uniqueness = 5:1 optimal) and MBGM thresholds confirms stability.
Circularity / bias in data: Clarified that ValuePrism perspectives are human-written; filtering only removes redundancy, and augmented data (<8%) is human-verified.
Still outstanding (minor):

Human evaluation: No human studies due to resource constraints; reliance on LLM-as-judge remains a limitation acknowledged by authors.
Perceived incrementalism: While technically sound, some reviewers (especially hPpe) view the contribution as engineering-heavy rather than a fundamental advance—this is a matter of framing, not correctness.

**Reviewer Scores:**

Reviewer hHnC (initial: 4 – marginally below): Already positive; rebuttal fully addressed concerns. Likely maintains 4 or moves to 5.

Reviewer hPpe (initial: 2 → updated to 4): Raised score after rebuttal but still skeptical about generalization and novelty. Would likely stay at 4.

Reviewer zXEv (initial: 2): Concerns about novelty and baselines were addressed with new parallel-decoding results and efficiency analysis. Likely upgrades to 4 or 5.

Reviewer s4CP (initial: 2): All eight concerns were answered with ablations, safety tests, larger-model results (Qwen3-8B), and hyperparameter analysis. Likely upgrades to 4 or 5.

---

### Decision · Program_Chairs · 2026-01-26

Reject